Identification of Dioscorea opposite Thunb. CDPK gene family reveals that DoCDPK20 is related to heat resistance

Gao Yuanli
Zhang Yanfang
Ji Xiang
Wang Jinxin
Suo Ningning
Liu Jiecai jiecailiu@163.com
Huo Xiuwen huoxiuwen@imau.edu.cn
Inner Mongolia Agricultural University , Hohhot , China
Kumar Ravinder
Electronic publication date: 2023 Sep 20
Publication date: 2023
Volume: 11
Electronic Location ID: e16110
Received 2023 Mar 8; Accepted 2023 Aug 25
Copyright: © 2023 Gao et al.
Copyright year: 2023
Copyright holder: Gao et al.
License: This is an open access article distributed under the terms of the Creative Commons Attribution License, which permits unrestricted use, distribution, reproduction and adaptation in any medium and for any purpose provided that it is properly attributed. For attribution, the original author(s), title, publication source (PeerJ) and either DOI or URL of the article must be cited.
License URL: https://creativecommons.org/licenses/by/4.0/

Keywords: CDPK gene family, Dioscorea opposite Thunb., Expression pattern, Heat temperature, DoCDPK20, Transgenic tobacco

Funding: National Natural Science Foundation of China 31860558 Science and Technology Program of Inner Mongolia Autonomous Region 2020GG0044 This study was co-funded by the National Natural Science Foundation of China (31860558) and the Science and Technology Program of Inner Mongolia Autonomous Region (2020GG0044). The funders had no role in study design, data collection and analysis, decision to publish, or preparation of the manuscript.

==============================
Temperature affects the growth and yield of yam (Dioscorea opposite Thunb.), and calcium-dependent protein kinases (CDPKs) play an important role in the plant stress response. However, there has been a lack of system analyses of yam’s CDPK gene family. In this study, 29 CDPK transcriptome sequences with complete open reading frames (ORFs) were identified from yam RNA sequencing data. The sequences were classified into four groups (I–VI) using phylogenetic analysis. Two DoCDPK genes were randomly selected from each group and the gene patterns of yam leaves were determined using quantitative real-time PCR (qRT-PCR) under high and low temperature stress in order to show their unique functions in mediating specific responses. Among them, DoCDPK20 was significantly induced in high temperatures. The pPZP221-DoCDPK20 was transformed into tobacco leaves using an agrobacterium-mediated method. Under high temperature stress, DoCDPK20 overexpression reduced photosynthesis and improved heat tolerance in transgenic tobacco. Our research offers meaningful perspectives into CDPK genes and new avenues for the genetic engineering and molecular breeding of yam.

Introduction

Yam (Dioscorea opposite Thunb.) is widely distributed in Africa, Asia, Oceania, and South America and plays significant economic, medicinal, and socio-cultural roles (Kwabena et al., 2020; Easter et al., 2022). In global production, yam ranks fourth among root crops, after potato, cassava, and sweet potato (Chandrasekaran et al., 2016; Sharif et al., 2020). The best growing temperature for yams is between 25–35 °C (Onwueme, 1977). Due to global warming (Mao et al., 2017; Guwei et al., 2021), climate anomalies have become one of the major problems in crop cultivation (Ravi et al., 2011; Kumar et al., 2016). High temperature can damage yield, yield traits, plant growth, and productivity (Zlatev & Lidon, 2012). In particular, yam yield decreases with increasing temperature (More, Ravi & Raju, 2017). High temperature not only causes significant external damage to plants, but also affects cell membrane structure, osmotic regulation mechanisms, antioxidant systems, and photosynthesis (Foyer & Noctor, 2010; Lobell & Field, 2007; Ohama et al., 2017; Tian & Yao, 2011). Plants have evolved to alleviate the damage of high temperatures through self-regulation, such as accumulating large amounts of proline (Pro) and soluble protein (Sonal, Divya & Anjana, 2014). These processes can increase the osmotic pressure of plants, which helps them maintain sufficient water and alleviate transpiration caused by high temperatures (Sonal, Divya & Anjana, 2014). Although plants have multiple ways of regulating and dissipating excess energy (Javier, Néstor & Estela, 1999), the process of excess energy dissipation inevitably leads to the accumulation of reactive oxygen species (ROS) and hydrogen peroxide substances (Javier, Néstor & Estela, 1999). Therefore, improving the heat resistance of plants has become an important issue for researchers and traditional crop breeders.

Calcium-dependent protein kinases (CDPKs) are widely found in a variety of plants and play an important role in the calcium signaling pathway, which has the largest family of CDPK genes (Ray et al., 2007; Crizel et al., 2020). CDPKs are in the upstream phase of the calcium signaling pathway, and they can regulate a large number of genes and metabolic processes with a wide range of functions (Sung et al., 2003). Numerous studies have shown that CDPK genes play an important role in plant growth, development, and stress responses, including heat tolerance and photosynthesis (Hamel, Sheen & Séguin, 2014; Romeis & Herde, 2014; Simeunovic et al., 2016; Zhu et al., 2021). Plants with high ZmCDPK7 expression showed higher temperature tolerance and photosynthesis synthesis rates in maize (Zhao et al., 2021). Researchers showed that overexpression of the tomato LeCPK2 gene improved the heat tolerance of transgenic tobacco (Chang et al., 2011). SlCDPK16 gene in tomato has been shown to be associated with heat stress (Hu et al., 2016). CDPK proteins play a crucial role in high-temperature signal transduction of cells. However, there has been little research on CDPKs in yams and their role in high temperature stress response.

This study was based on a search of yam transcriptomics data, and we analyzed the response of the CDPK gene family to temperature stress. The overexpression of a key heat-resistant gene confirmed its characteristics and molecular functions associated with heat resistance. These results will form the basis for future molecular studies and genetic engineering of heat resistance in yam.

Materials and Methods

Identification of CDPK genes, phylogenetic analysis, and sequence analyses of Dioscorea opposite Thunb•

To obtain the sequences of the Dioscorea opposite Thunb. CDPK gene family, we searched from pre-transcriptomic sequencing of yam tubers (Gao et al., 2022). The Pfam database was used to acquire the Hidden Markov Model (HMM) file (PF00069) for the CDPK domain (Table 1) (Punta et al., 2004). Then, the yam transcriptome was compared to BLASTp using HMMER 3.0 (E-value set to 0.01) of the model as a sample. For the validation of the candidate gene CDPK, we used the online program PFAM, which contained Conserved Domains Database Search and SMART tool (Table 1). Prediction of open reading frames (ORFs) was performed via the website (https://www.ncbi.nlm.nih.gov/gorf/gorf.html) and follow-up tests.

Table 1 Online tools and databases used in this study.

Online tools and databases	URL	
The Pfam database	https://www.ebi.ac.uk/interpro	
PFAM	http://pfam.xfam.org/search	
Conserved domains data base	https://www.ncbi.nlm.nih.gov/Structure/cdd/wrpsb.cgi	
SMART tool	http://smart.embl-heidelberg.de/	
The MEME	https://meme-suite.org/meme/	
InterPro scan	http://www.ebi.ac.uk/interpro/	
ITOL	http://itol.embl.de/help.cgi	

We used the online ExPasy program (http://www.expasy.org/tools/) to predict the physical properties of CDPK members, such as length, molecular weight (MW), and isoelectric point (pI) (Wang et al., 2019). We used an online tool, the MEME website, to determine the motif, to annotate it with InterPro Scan, set the parameters to 0 or 1 occurrence per sequence (up to 10 subjects), and set the length of each motif to 5–50 aa (Table 1) (Bailey et al., 2015).

The amino acid sequence of DoCDPK of the ORF was compared with the rice CDPK and we used the Clustal Omega tool to perform multiple sequence alignment (Table 1). According to the above results, we used the maximum likelihood of MEGAX to construct a phylogenetic tree with 1,000 bootstrap replications of maximum likelihood (Tamura et al., 2011) and used ITOL to draw (Table 1).

Plant materials and growth conditions

The yam (Dioscorea opposite Thunb.) cultivar Dahechangyv was used as plant material. First, yam sterile seedlings were obtained from young stem tips of plants that were grown in the field for 2 months. They were rinsed under running water for 30 min, disinfected with 75% alcohol for 30 s, soaked with 0.1% Hgcl2 for 8 min, and washed with sterile water three to four times. Next, these explants were inoculated in MS medium. Then, these sterile seedlings were placed in a lighted incubator (25 °C, 2000 Lux, 16 h of light, 8 h of dark incubation). Using previous methods as references (Weckwerth, Ehlert & Romeis, 2014; Tian et al., 2016; Chang et al., 2011), yam sterile seedlings were selected for uniform growth and placed in either a −4 °C low temperature incubator or a 42 °C incubator. We collected yam leaves at 0, 1, 2, 4, 8, and 12 h; snap-froze them in liquid nitrogen; and stored them at −80 °C for gene expression analysis.

Analysis of CDPK gene expression

The RNA of all samples were extracted using Trizol (TIANgel, Beijing, China) according to the manufacturer’s descriptions. The quality of the RNA samples were tested using NanoDrop2000c (Thermo Fisher Scientific, Waltham, MA, USA) and gel electrophoresis. BiomarkerScript III RT Master Mix (Biomarker, Beijing, China) was used to perform reverse transcription. We used the TB Green Premix Ex Taq II kit (TaKaRa, Dalian, China) to implement qRT-PCR on the FTC-3000P (Funglyn Biotech, Richmond, ON, Canada). The 18S-rRNA gene is an actin gene. Primer Premier 5.0 software was used to design the primers based on transcriptomics sequencing data. The primers and reaction procedure are shown in Table S1. The experiment was set up with three replications. All qRT-PCR data were calculated using 2−ΔΔCt.

Cloning DoCDPK20 and constructing pPZP221-DoCDPK20 plant expression vector

PCR amplification was performed using ORF special primers of DoCDPK20 (Table S1). The ORF of DoCDPK20 was connected into the pGM-T vector (PGM-T Vector Linking Kit; TaKaRa, Dalian, China) and sequenced for confirmation. The ORF of DoCDPK20 was amplified using the specific primers DoCDPK20-Z with restriction enzyme cleavage sites (Kpn I and BamH I), and the amplification product was connected into the pPZP221-35S-NOS vector. Table S1 contains the primer sequences. The pPZP221-DoCDPK20 plant expression vector was obtained, and Agrobacterium tumefaciens EHA101 (Weidi Biotechnology, Shanghai, China) was transformed using the freeze-thaw method.

Preparation of transgenic tobacco plants that overexpressing DoCDPK20

The Agrobacterium-mediated transformation method was used to transform wild-type tobacco (Nicotiana tabacum L., labeled WT) with Agrobacterium containing the plant expression vector pPZP221-DoCDPK20. Positive seedlings were screened for resistance using MS medium (containing 6-BA 1.0 mg/L, gentamicin 50 mg/L, and Cefotaxim 500 mg/L). Transgenic plants were grown on 1/2 MS medium (containing gentamicin 50 mg/L and Cefotaxim 500 mg/L). RNA and DNA assays were performed using the Hi-DNAsecure Plant Kit (TIANGEN, Beijing, China). Transgenic tobacco plants with high expression (labeled T2, T3, and T7) were selected for the next experiments.

Phenotypic observation and photosynthesis-related and physiological index measurements of transgenic tobacco

Transgenic and WT tobacco plants were washed from the culture medium and transferred to the substrate for 2 weeks. For follow-up experiments, we selected transgenic and WT tobacco plants with uniform growth. Under normal growth conditions, transgenic and WT tobacco plants were used as the control group. Under high temperature stress for 8 h, these plants were used as the experimental group. The morphological changes of the transgenic and WT tobacco plants were photographed and recorded at each stage. We selected plants with inverted trifoliate leaves (mature leaves) and used a photosynthesis meter (CIRAS-3, Hansatech, Norfolk, UK) to measure the gas exchange parameters in fully unfolded leaves, including photosynthetic rate (Pn), intercellular CO2 concentration (Ci), stomatal conductance (Gs), and transpiration rate (Tr). Pn/Tr is the water use efficiency (WUE) (Nieva et al., 1999). Fast chlorophyll fluorescence induction curves were measured using a continuous excitation fluorometer (M-PEA, Hansatech, Norfolk, UK) and set up with five replicates. The first true leaves were collected for histochemical staining observation. Under high temperature treatment, hydrogen peroxide (H2O2) and superoxide anion (O2−) levels were tested using 3,3-diaminobenzidine (DAB) and nitroblue tetrazolium chloride (NBT) colouring, respectively (Zhao, Liu & Xu, 2009; Hans, Zhang & Wei, 1997). The fully expanded inverted third and fourth blades were sampled, snap-frozen in liquid nitrogen, and stored at −80 °C for subsequent experiments. Peroxidase (POD), superoxide dismutase (SOD) and catalase (CAT) activity, and malondialdehyde (MDA), proline (Pro) and soluble protein content, were measured, respectively, and the experiment was set up with three biological replicates (Li, 2000).

Statistical analysis

The gene expression, photosynthetic parameter, and physiological indicator data were tested at the p < 0.05 level. Data statistics were conducted using IBM SPSS Statistics 25 (IBM, Armonk, NY, USA) and Excel 2010 (Microsoft Corporation, Redmond, WA, USA).

Results

Identification of DoCDPKs in Dioscorea opposite Thunb

Twenty-nine DoCDPKs were obtained from Dioscorea opposite Thunb. and labeled DoCDPK1-DoCDPK29 based on blastp comparison of CDPK protein sequences from other plants. Among the 29 DoCDPKs, the cDNA length of ORFs were predicted to range from 930 bp (DoCDPK29) to 1,827 bp (DoCDPK5). To further investigate the characteristics of DoCDPKs with complete sequences, the pI and MW of 29 DoCDPK proteins were calculated. The pI ranged from 5.5 (DoCDPK1) to 8.6 (DoCDPK22) and MW ranged from 35.11 kDa (DoCDPK29) to 67.67 kDa (DoCDPK5) (Table 2).

Table 2 The statistic information of DoCDPKs in Dioscorea opposite Thunb.

Gene ID	Gene name	ORF (bp)	PI	MW
(kDa)	Subcellular
localization	Clusters	
transcript_HQ_D_transcript21740	DoCDPK1	1,629	5.5	60.4	Plasma membrane	GroupI	
transcript_HQ_D_transcript18003	DoCDPK2	1,677	5.62	61.86	Plasma membrane	GroupI	
transcript_HQ_D_transcript18607	DoCDPK3	1,701	5.62	61.47	Plasma membrane	GroupI	
transcript_HQ_D_transcript16242	DoCDPK4	1,713	5.17	62.55	Plasma membrane	GroupI	
transcript_HQ_D_transcript12177	DoCDPK5	1,827	5.76	67.67	Plasma membrane	GroupI	
transcript_HQ_D_transcript17629	DoCDPK6	1,506	5.29	56.27	Plasma membrane	GroupI	
transcript_HQ_D_transcript19341	DoCDPK7	1,566	5.77	58.5	Plasma membrane	GroupII	
transcript_HQ_D_transcript16812	DoCDPK8	1,392	8.58	51.73	Plasma membrane	GroupII	
transcript_HQ_D_transcript17700	DoCDPK9	1,461	6.9	54.34	Plasma membrane	GroupII	
transcript_HQ_D_transcript16354	DoCDPK10	1,632	6.07	60.65	Plasma membrane	GroupII	
transcript_HQ_D_transcript19730	DoCDPK11	1,608	6.24	60.28	Plasma membrane	GroupIII	
transcript_HQ_D_transcript17708	DoCDPK12	1,596	6.52	59.75	Plasma membrane	GroupIII	
transcript_HQ_D_transcript17941	DoCDPK13	1,596	6.52	59.78	Plasma membrane	GroupIII	
transcript_HQ_D_transcript17896	DoCDPK14	1,584	6.46	59.53	Plasma membrane	GroupIII	
transcript_HQ_D_transcript18755	DoCDPK15	1,707	5.68	63.79	Plasma membrane	GroupIII	
transcript_HQ_D_transcript29156	DoCDPK16	1,323	6.76	49.12	Plasma membrane	GroupIII	
transcript_HQ_D_transcript20105	DoCDPK17	1,761	5.9	65.69	Plasma membrane	GroupIII	
transcript_HQ_D_transcript18594	DoCDPK18	1,623	6.13	60.53	Plasma membrane	GroupII	
transcript_HQ_D_transcript19307	DoCDPK19	1,605	5.13	60.03	Plasma membrane	GroupIII	
transcript_HQ_D_transcript18246	DoCDPK20	1,059	7.07	39.94	Plasma membrane	GroupIV	
transcript_HQ_D_transcript18497	DoCDPK21	1,530	8.31	57.33	Plasma membrane	GroupIV	
transcript_HQ_D_transcript14404	DoCDPK22	1,797	8.6	66.14	Extracellular	GroupIV	
transcript_HQ_D_transcript12339	DoCDPK23	1,749	8.03	65.11	Extracellular	GroupIV	
transcript_HQ_D_transcript18760	DoCDPK24	1,083	5.58	40.96	Plasma membrane	GroupIII	
transcript_HQ_D_transcript9433	DoCDPK25	1,071	5.99	40.56	Plasma membrane	GroupIII	
transcript_HQ_D_transcript11733	DoCDPK26	945	7.67	35.95	Plasma membrane	GroupIV	
transcript_HQ_D_transcript18398	DoCDPK27	1,587	8.47	59.25	Plasma membrane	GroupIV	
transcript_HQ_D_transcript15074	DoCDPK28	1,062	6.78	40.24	Plasma membrane	GroupIV	
transcript_HQ_D_transcript8194	DoCDPK29	930	5.71	35.11	Plasma membrane	GroupIII	

Phylogenetic analysis of DoCDPKs

To understand the phylogenetic relationships of CDPK proteins, the sequences of CDPK proteins in rice (Oryza sativa) and yam (Dioscorea opposite Thunb.) were used to construct phylogenetic trees. All CDPKs were divided into four groups. Group I included 11 OsCDPKs and six DoCDPKs. Group II contained eight OsCDPKs and five DoCDPKs. Group III consisted of eight OsCDPKs and 11 DoCDPKs. Group IV was comprised of two OsCDPKs and seven DoCDPKs (Fig. 1). To further analyze the structure of the DoCDPK proteins, we identified their conserved motifs based on the MEME databases. Ten conserved motifs were identified (named Motif 1–10), length between 21 and 50 amino acids (Fig. 2). The Group I proteins contained motifs 7, 5, 4, 1, 2, 3, 6, 9, and 8; Group II proteins contained motifs 7, 5, 4, 1, 2, and 3; Group III proteins contained motifs 2 and 3; and Group IV proteins contained motifs 4, 1, 2, 3, and 6 (Fig. 2A). Notably, each DoCDPK contained five to 10 conserved motifs, and each gene sequence contained motifs 2 and 3.

Figure 1 Phylogenetic tree of CDPK family from Oryza sativa L. and Dioscorea opposite Thunb.

The clusters were designated as group I–IV and indicated in a specific color.

Figure 2 (A–B) Motif analysis of DoCDPKs in Dioscorea opposite Thunb.

The DoCDPK gene expression pattern of temperature stresses

To investigate the possible role of DoCDPK under low (−4 °C) and high (42 °C) temperature stress, we performed qRT-PCR assay in yam leaves. We randomly selected two genes from each group and monitored these gene expression trends (Fig. 3). DoCDPK1 and DoCDPK6 were selected from Group I, DoCDPK7 and DoCDPK10 from Group II; DoCDPK16 and DoCDPK17 from Group III, and DoCDPK20 and DoCDPK22 from Group IV. Under low temperature stress, DoCDPK1, 6, and 7 were down-regulated compared with the control (0 h), and DoCDPK16 was opposite (Fig. 3A). The remaining four DoCDPK genes were responsive to low temperature stimulation and down-regulated after 1 h of treatment (Fig. 3A). Among them, DoCDPK10 and DoCDPK20 increased 4.5 and 6.9-fold at 12 and 8 h of low temperature treatment, respectively, compared to the control (0 h) (Fig. 3A). Genes in Group I were significantly repressed under low-temperature stress. DoCDPK7 and DoCDPK10 in Group II were significantly repressed and induced under low temperature stress, respectively. The selected genes in Group III and Group IV were both significantly induced to be highly expressed (Fig. 3A). Under high temperature stress, DoCDPK6, 7, 10, 16, and 22 were induced to express higher values at 1, 4, 4, 8 and 4 h, respectively (Fig. 3B). Similarly, under low temperature stress, DoCDPK1 expression was inhibited (Fig. 3B). DoCDPK17 and DoCDPK20 gene expression levels were similarly down-regulated at 1 h, the former reaching a high value at 4 h and 3.9 times higher than the control (0 h), and the latter reaching a high value at 8 h and 8.4 times higher than the control (0 h) (Fig. 3B). DoCDPK1 and DoCDPK6 of Group I were significantly repressed and induced under high temperature stress, respectively. The genes in Group II, Group III, and Group IV were significantly induced (Fig. 3B). However, the changes of gene expression were more pronounced in Group IV when compared to the other groups (Fig. 3). In summary, DoCDPK20 showed the maximum change in the level of expression at 8 h after high temperature stress (Fig. 3). Thus, we speculated that DoCDPK20 may be a significant gene in high temperature stress response and conducted the following functional analysis.

Figure 3 Based on qRT-PCR assay expression patterns of eight DoCDPK genes under low temperature (A) and high temperature (B) stress.

The letters of the upper left corner represent gene names in the figure; the lowercase letters in the figure represent p < 0.05 significance analysis of variance.

Phenotypic changes in DoCDPK20 transgenic tobacco under high temperature stress

To determine whether the gene enhances the heat tolerance of plants, we used by the pPZP221 plant expression vector to expand the ORF cDNA of DoCDPK20 from yam tuber and overexpress in Nicotiana tabacum L. We obtained eight transgenic tobacco strains and selected three transgenic strains (T2, T3, and T7) to characterize the heat-induced phenotypes. Overexpression of DoCDPK20 notably improved the high temperature tolerance in tobacco (Fig. 4). Under normal growth conditions, there was no significant difference in morphology between WT and overexpression of DoCDPK20 transgenic tobacco. In plants overexpressing DoCDPK20, the newly developed leaves of the transgenic tobacco grew normally, while the WT foliage crumpled and rolled up after 8 h of high temperature treatment. These results indicate that the transgenic tobacco exhibited healthier morphological characteristics than WT tobacco plants under high temperature treatment.

Figure 4 Overexpression of DoCDPK20 enhances the heat tolerance of transgenic tobacco plants.

Observation of phenotypic changes in wild-type (WT) and transgenic (T2, T3 and T7) tobacco plants at 0 and 8 h under the high temperature.

Effect of heat treatment on photosynthesis-related indicators in DoCDPK20-overexpressing plants

To investigate the causes of phenotypic changes in transgenic tobacco and WT plants, we performed photosynthesis analysis under thermal stress. After 8 h of high temperature treatment, Pn, Gs, Ci, and Tr were significantly higher in overexpressed DoCDPK20 plants than WT plants. After heat treatment, WUE was significantly higher in WT than transgenic tobacco (Fig. 5A). After 8 h of high temperature treatment, Pn, Gs, Ci, and Tr in transgenic tobacco increased compared with WT tobacco. The Tr of transgenic tobacco was 1.4 times higher than WT, and the WUE of transgenic tobacco was 0.8 times lower than WT tobacco. Under high temperature treatment, the rapid chlorophyll fluorescence induction kinetic curves of DoCDPK20 overexpression and WT tobacco leaves showed a typical 0-J-I-P curve. Fluorescence rose from the lowest level (Fo or O) to the highest level (Fm or P) in three different stages, and the two points were named J and I, respectively.

Figure 5 Overexpression of DoCDPK20 affect photosynthesis-related indicators in transgenic tobacco plants of high temperature.

Changes in photosynthetic parameters (A), OJIP (B) and fluorescence parameters (C) of transgenic tobacco and WT tobacco; the lowercase letters in the figure represent p < 0.05 significance analysis of variance.

After high-temperature stress, the I and P points of the leaves showed a significant decrease compared with the control, and the difference was more pronounced in WT than transgenic tobacco plants (Fig. 5B). As the heat stress prolonged, the maximum photochemical efficiency (Fv/Fm) significantly decreased, but there was no significant difference between transgenic tobacco and WT at different treatment stages. PIabs is a comprehensive parameter that reflects plant photosynthetic performance. Under high temperature treatment, PIabs was significantly lower than the control and significantly higher in transgenic tobacco than WT tobacco, with transgenic tobacco PIabs elevated 1.2-fold compared to WT tobacco (Fig. 5C). These results suggest that overexpression of DoCDPK20 improved the heat tolerance of transgenic tobacco.

Changes in physiological indicators of heat tolerance in DoCDPK20-overexpressed plants

In order to further analyze the causes of phenotypic changes in transgenic tobacco and WT plants, DAB and NBT staining were performed at 0 and 8 h under heat treatment, respectively. For DAB staining, the foliage of both transgenic tobacco and WT plants started to colour after 8 h of heat treatment, with WT darkening more compared to the transgenic leaves. These results suggest that WT tobacco accumulated more H2O2 than transgenic tobacco leaves (Fig. 6A). For NBT staining, the leaves of transgenic tobacco and WT plants started to colour after 8 h of heat treatment, with the WT becoming significantly darker than the transgenic tobacco leaves. These results indicate that WT tobacco accumulated more O2− (Fig. 6A). In summary, DoCDPK20 overexpression delayed the accumulation of H2O2 and O2− in the leaves. This may be because CDPK enhances the resistance of plants and protects them from damage caused by accumulation of peroxides and active oxygen.

Figure 6 Overexpression of DoCDPK20 enhances high temperature stress in transgenic tobacco.

(A) DAB and NBT staining of transgenic and WT tobacco. (B) High temperatures affect physiological indicators in transgenic and WT tobacco.

To better understand the differences between the overexpression of DoCDPK20 and WT plants, the activities of SOD, POD, and CAT, and the contents of soluble sugar, Pro, and MDA were tested under high-temperature stress. SOD, POD, and CAT are known as protective enzyme systems that can scavenge ROS and other superoxide radicals from plants. Our results showed that the activities of SOD, POD, and CAT showed the same trend. Under normal growth conditions, there was no difference in the enzyme activities of transgenic and WT tobacco plants. After 8 h of heat treatment, the enzyme activity of the transgenic tobacco increased significantly, and was significantly higher than the control (Fig. 6B). The activities of SOD, POD, and CAT in transgenic tobacco were 1.1, 1.6, and 1.2 times higher than WT, respectively, at 8 h of high temperature treatment. Compared with the control, the content of soluble sugar and Pro in transgenic tobacco and WT plants increased under heat treatment, and transgenic tobacco plants had significantly higher levels than WT (Fig. 6B). The soluble sugar and Pro content of the transgenic tobacco was higher than that of the WT tobacco at 8 h of high temperature treatment. The MDA content significantly increased in transgenic and WT tobacco plants under heat treatment, and WT tobacco plants had significantly higher levels compared to transgenic tobacco. The MDA content of transgenic tobacco was 0.6 times lower than that of WT at 8 h of high temperature treatment. These results demonstrated that the increase of SOD and POD activity protected plant cell membrane from high temperature damage. The increase of soluble sugar and Pro content increased the concentration of plant cytosol. Compared to transgenic tobacco, WT accumulated more H2O2 and MDA, which can cause more damage to plants. In summary, the high temperature resistance of transgenic tobacco plants improved.

Changes in DoCDPK20 gene expression in transgenic tobacco for heat tolerance

To further investigate the mechanism of heat resistance of DoCDPK20 overexpression, we measured DoCDPK20 gene expression in transgenic tobacco plants. Our results illustrate that DoCDPK20 gene expression was significantly elevated under heat treatment conditions. The overexpression of DoCDPK20 plants was significantly higher than WT tobacco at different stages (Fig. 7). The expression of DoCDPK20 gene was 7.2-fold higher in WT and 12.0-fold higher in transgenic tobacco compared to the control (0 h). At 8 h of high temperature treatment, the expression of DoCDPK20 gene in transgenic tobacco was 1.7-fold higher than in WT tobacco (Fig. 7). These results suggest that DoCDPK20 plays an important role in high temperature conditions.

Figure 7 The DoCDPK20 gene in transgenic tobacco is induced to be highly expressed under high temperature conditions. The lowercase letters in the figure represent p < 0.05 significance analysis of variance.

Discussion

The CDPK gene family is widely found in all green plants (Wan, Lin & Mou, 2007) and has very ancient origins (Chen et al., 2013). Advances in the genetic identification and functional evaluation of CDPK family members have revealed that these genes play a significant role in plant responses to multiple abiotic stresses (Ranty et al., 2016). Studying the CDPK gene family will contribute to the evolution and function of specific CDPK gene developments and stress responses in yam. Thirty-four, 31, 29, 25, and 29 CDPK genes have been identified in Arabidopsis, rice, tomato, oilseed rape, and poplar, respectively (Hrabak et al., 2003; Ray et al., 2007; Hu et al., 2016; Valmonte et al., 2014; Zhang et al., 2014). In this study, 29 CDPK genes of yam were identified. This suggests that the DoCDPK gene family in yam is evolutionarily conserved in terms of gene number. From bryophytes to angiosperms, the CDPK gene family has exhibited a high degree of structural conservation over the course of its evolution (Harmon, Gribskov & Harper, 2000). The gene family is essentially divided into four subclasses (Wan, Lin & Mou, 2007). Our research findings suggest that DoCDPK is also divided into four groups and the classification of DoCDPK is also conserved. Additionally, there are 10 different motifs across the different DoCDPK proteins, each protein contains five to 10 conserved motifs, and motifs 2 and 3 are common to all groups of genes.

CDPKs play an important role in plant growth, development, and response to most external stress (Ranty et al., 2016). There is abundant evidence that high temperatures can induce specific expression of CDPK genes in many plants. In mountain grapes, high temperatures induced the expression of VaCPK9, VaCPK20, VaCPK21, and VaCPK29 genes (Dubrovina, Kiselev & Khristenko, 2013). Heat treatment increased the expression of the MsCPK3 gene in cultured alfalfa cells and OsCPK25 gene in rice seedlings (Geng et al., 2011; Wan, Lin & Mou, 2007). In our experiments, DoCDPK20 was significantly highly expressed in yam after high temperature treatments, which was consistent with the results of previous studies on mountain grapes (Dubrovina et al., 2015). In our experiments, we cloned and overexpressed DoCDPK20 of CDPK gene family Group IV in tobacco and found that overexpression of this gene significantly increased the heat stress of transgenic tobacco plants. Short or prolonged periods of high temperatures can cause a range of physiological and biochemical changes in plants. Photosynthesis is one of the physiological processes in plants that is most sensitive to high temperatures (Allen & Ort, 2001). In this experiment, Pn, Gs, Ci, and Tr were significantly reduced under high temperature stress but significantly higher in transgenic tobacco than WT. WUE was significantly higher under high temperature stress and the overexpression of DoCDPK20 plants was significantly lower than WT. Our research findings suggest that plant photosynthesis decreases under heat stress (Andersson & Backlund, 2008) and overexpression of the DoCDPK20 gene alleviates the damage to photosynthesis in transgenic tobacco plants under heat treatment. This may be due to stomatal limitation during this treatment period, when stomatal conductance is reduced and CO2 availability is inadequate resulting in a decrease in photosynthesis (Cui et al., 2016). Stomatal limitation gradually decreased with the prolongation of high temperature stress, and Pn also decreased. At this time high temperatures did not cause damage to the internal physiology of the plant, and when the high temperature was lifted, the stomata returned to normal levels which, to a certain extent, reflected the plant’s heat tolerance (Hamerly & Knapp, 1996). High temperatures affect not only the degree of stomatal opening and closing, but also the photosynthetic reaction centre PSII (White & Critchley, 1999). Chlorophyll fluorescence induction kinetic curves can reflect the primary photochemical reactions of the PSII centre (Guo et al., 2020). PSII adapts to high temperatures by regulating the opening of reaction centres and photochemical quantum transfer, and maintaining CO2 assimilation capacity while reducing the damage to itself by increasing heat dissipation (Sadura & Janeczko, 2022). In this study, the fluorescence intensity in the O-J phase of tobacco (including transgenic tobacco and WT) was higher under high temperature than that of tobacco under normal growing conditions (including the overexpression of DoCDPK20 and WT plants), indicating that the leaves received a large number of light quanta (Camejo et al., 2005). During electron transfer, the fluorescence parameters (J-I and I-P phases) were lower in tobacco (both transgenic and WT) under high temperatures than under normal growth conditions and were higher in the transgenic tobacco plants than WT, indicating a reversible inactivation of the PSII centre. This allows more excess light energy to be dissipated as heat in transgenic tobacco than in WT under high temperature treatment. PIabs was significantly lower under high temperature treatment and the overexpression of DoCDPK20 plants was significantly higher than in WT. Our research findings indicate that overexpression of DoCDPK20 alleviated the damage of heatt treatment on the photochemical activity of transgenic tobacco leaves and improved the photosynthetic performance as well as stress tolerance of the transgenic plants.

Under heat treatment, plant cells are damaged. Soluble protein and Pro in plants are significant osmoregulation substances, and their elevated levels can increase cytosol concentration, decrease cell permeation potential, and reduce the damage to leaves under adverse conditions (Duran-Serantes, Gonzalez & Reigosa, 2002; Liang et al., 2018). In our study, three DoCDPK20 transgenic tobacco strains (T2, T3, and T7) with DoCDPK20 overexpression, showed a healthier phenotypic morphology under high temperature stress with higher soluble protein and Pro content than WT. MDA content can respond to the degree of peroxidation of the cell membrane mass (Guo et al., 2020). ROS content can respond to the extent of damage to plant cell membranes, and the antioxidant enzyme system is involved in the scavenging of ROS (Ahmad et al., 2016). H2O2 can be catalyzed by POD and CAT to phenols or amines to prevent free radical damage to cells (Ueda et al., 2013; Bowler, Montagu & Inze, 1992). In this study, the staining results for DAB and NBT demonstrated that overexpression of DoCDPK20 plants led to decreased accumulation of H2O2 and O2 compared to WT. SOD, CAT, and POD activity of the overexpressed DoCDPK20 tobacco plants were significantly higher than those of the WT under high temperature treatment. In contrast, transgenic tobacco MDA content was significantly lower than that of WT.

Ca2+ is an important intracellular signaling molecule that binds to receptor proteins and regulates many physiological processes (Sanders et al., 2002). CDPK is an important multifunctional receptor protein for Ca2+ in organism cells that can transduce Ca2+ information to downstream target proteins. It is an important member of the Ca2+ signal transduction pathway (Couto & Zipfel, 2016). When cells are stimulated to increase cytoplasmic Ca2+ concentration, CDPK binds to Ca2+ and forms active calcium-dependent protein kinase (Harmon, Gribskov & Harper, 2000). These results lead to a series of physiological effects in the body (Harmon, Gribskov & Harper, 2000). In this study, we hypothesized that the expression of DoCDPK20 was induced in transgenic tobacco under high temperature stimulation. DoCDPK20 is the main gene that controls CDPK activity under this treatment condition. Under high temperature treatment, Ca2+ concentration and CDPK activity are increased and combined. They form an active CDPK and interact with downstream target proteins. It is possible that these processes lead to a series of physiological responses in the plant to resist high temperatures. In summary, our experiments demonstrate that the overexpression of DoCDPK20 increases the thermal stress of transgenic tobacco.

Conclusion

In summary, we identified a total of 29 DoCDPKs with complete ORFs in yam, and conducted a comprehensive and systematic analysis on these genes. These genes were classified into four groups, which were analysed using gene expression and screened for key DoCDPK members against extreme temperatures. Among them, DoCDPK20 played an important role under high temperature stress. In order to further validate the function of the DoCDPK20 gene, the gene was overexpressed in WT tobacco, and overexpression of the DoCDPK20 gene significantly improved the heat tolerance of transgenic tobacco. DoCDPK20 can be used when studying heat tolerance and represents a possible genetic resource that could be beneficial for improving heat resistance in plants.

Supplemental Information

Supplemental Information 1 Raw data.

Physiological indicators and photosynthesis-related indicators.

Click here for additional data file.

Supplemental Information 2 The primers used in this study.

Note: GGATCC and GTCGAC are the restriction sites of Kpn I and BamH I

Click here for additional data file.

Additional Information and Declarations

Competing Interests

Author Contributions

Data Availability

The authors have no competing interests to declare.

Yuanli Gao conceived and designed the experiments, performed the experiments, analyzed the data, prepared figures and/or tables, authored or reviewed drafts of the article, and approved the final draft.

Yanfang Zhang conceived and designed the experiments, performed the experiments, analyzed the data, authored or reviewed drafts of the article, and approved the final draft.

Xiang Ji conceived and designed the experiments, performed the experiments, authored or reviewed drafts of the article, and approved the final draft.

Jinxin Wang conceived and designed the experiments, performed the experiments, authored or reviewed drafts of the article, and approved the final draft.

Ningning Suo conceived and designed the experiments, performed the experiments, authored or reviewed drafts of the article, and approved the final draft.

Jiecai Liu conceived and designed the experiments, analyzed the data, authored or reviewed drafts of the article, and approved the final draft.

Xiuwen Huo conceived and designed the experiments, authored or reviewed drafts of the article, and approved the final draft.

The following information was supplied regarding data availability:

The raw data are available in the Supplemental Files.

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
