# Peer review of "Identification of Dioscorea opposite Thunb. CDPK gene family reveals that DoCDPK20 is related to heat resistance"

_PeerJ, doi:10.7717/peerj.16110_

## Round 0.1 · original submission · Major Revisions

This study presents an interesting investigation into the role of calcium-dependent protein kinases (CDPKs) in the response of yam to temperature stress. The use of RNA sequencing data to identify CDPK transcriptome sequences with complete Open reading frames (ORFs) and subsequent phylogenetic analysis to group them into six distinct groups is commendable.

The selection of two CDPK genes from each group and their quantification using quantitative real-time PCR (qRT-PCR) in yam leaves under high and low temperature stress provides valuable insights into their unique functions in mediating specific responses. Furthermore, the induction of DoCDPK20 in high temperature conditions is a significant finding that sheds light on its potential role in temperature stress response.

The use of the Agrobacterium-mediated method to genetically transform tobacco leaves using pPZP221-DoCDPK20 is a notable addition to the study, as it demonstrates the functional role of this CDPK in improving heat tolerance in transgenic tobacco plants.

However, it is important to note that the study only focuses on the response of yam to temperature stress and does not address other factors that may affect its growth and yield. Moreover, the use of tobacco leaves as a model system may not reflect the actual response of yam to temperature stress.

Overall, this study provides meaningful perspectives into the role of CDPK genes in the response of yam to temperature stress and offers a candidate for new avenues of genetic engineering and molecular breeding. However, further studies are needed to confirm the findings and to investigate the potential of these CDPK genes in other stress conditions that may affect yam growth and yield.

Reviewer 1 ·

Basic reporting

The manuscript “Identification of Dioscorea opposite Thunb. CDPK gene family reveals that DoCDPK20 is referred to heat resistance” deals with the role of CDPK gene family under high temperature stress in Dioscorea opposite Thunb. This is an interesting study that investigates the role of calcium-dependent protein kinases (CDPKs) in the growth and yield of yam, more specifically in how they react when they are subjected to temperature stress. In the study, RNA sequencing data are used to discover 29 different CDPK transcripts. These CDPK transcripts are then phylogenetically classified into six distinct categories. The authors then choose two DoCDPK genes from each group and utilize quantitative real-time PCR to evaluate the patterns of gene expression that occur in yam leaf tissue when the plant is subjected to high or low temperature stress.
he study has several strengths, including the use of RNA sequencing data to identify CDPK transcripts and the use of qRT-PCR to determine gene expression patterns in response to temperature stress. The use of the Agrobacterium-mediated method to genetically transform tobacco leaves with the selected gene is also a valuable approach.
However, there are some areas where the study could be improved. For example, the authors could provide more details about the methods used to select the DoCDPK genes from each group. Additionally, additional information about the mechanisms by which DoCDPK20 improves the heat tolerance of transgenic tobacco could be included, as this would help to provide a better understanding of the gene's function. This would be included because it would be helpful in providing a better understanding of the gene's function.

Comments
• In the abstract section, the author can include one statement on the future impact of the present study
• LN 43 and 45: The author should maintain the consistency of the gene format mentioned. All gene names should be in italics.
• The introduction should be elaborated. The introduction section is too small.
• LN 89: Please rectify the spelling “lex”
• Ln 126-130: Please rewrite the line.
• Ln 145: Please elaborate on the section of “Statistical Analysis” section
• Ln 286-289: Please rewrite the line.
• The conclusion section needs to be revisited. It seems that the authors have hurriedly written this section.

Experimental design

Please see the basic reporting section

Validity of the findings

Please see the basic reporting section

Reviewer 2 ·

Basic reporting

Overall, the study provides interesting insights into the role of CDPK genes in yam's response to temperature stress and their potential application in genetic engineering and molecular breeding. However, there are some areas where the study could be improved:

Experimental design

The experimental design is appropriate.

Validity of the findings

Section "The DoCDPK Genes Expression Pattern of Temperature stresses" required more elaborative analysis within-group and between-group.

The study reports the overexpression of DoCDPK20 in tobacco and its effect on heat tolerance, but it does not provide any mechanistic insights into how DoCDPK20 achieves this effect. please elaborate on this aspect in the discussion part for better comprehension.

The study could benefit from more detailed descriptions of the phenotypic and physiological changes in the transgenic tobacco plants, and how these changes relate to the overexpression of DoCDPK20

---

## Round 0.2 · Minor Revisions

The authors have done modifications in the manuscript based on revieweres suggestions. These revisions were expected to enhance the overall quality of the document. However, it is crucial for the authors to carefully review the manuscript once again to ensure that it adheres to proper language standards and to correct any typographical errors that might have been overlooked.

Additionally, the authors should pay attention to any typographical errors present in the manuscript.

Reviewer 1 ·

Basic reporting

The authors made significant changes in the manuscript as per my suggestion. However, the author needs to recheck the manuscript for language checks and some typographical errors.

Experimental design

NA

Validity of the findings

NA

Additional comments

NA

Reviewer 2 ·

Basic reporting

The authors have revised the manuscript as per my suggestions and now can be accepted.

Experimental design

Very well done and presented

Validity of the findings

NA

Additional comments

NA

---

## Round 0.3 · accepted · Accept

The authors have revised the manuscript as per suggestions. Hence, now the manuscript can be accepted.

Congratulations.